# CD40 Signaling in Mice Elicits a Broad Antiviral Response Early during Acute Infection with RNA Viruses

**DOI:** 10.3390/v15061353

**Published:** 2023-06-12

**Authors:** Kai J. Rogers, Paige T. Richards, Zeb R. Zacharias, Laura L. Stunz, Rahul Vijay, Noah S. Butler, Kevin L. Legge, Gail A. Bishop, Wendy Maury

**Affiliations:** 1Department of Microbiology and Immunology, University of Iowa, Iowa City, IA 52242, USA; kai-rogers@uiowa.edu (K.J.R.); paige-richards@uiowa.edu (P.T.R.); laura-stunz@uiowa.edu (L.L.S.); rahul.vijay@rosalindfranklin.edu (R.V.); noah-butler@uiowa.edu (N.S.B.); kevin-legge@uiowa.edu (K.L.L.); gail-bishop@uiowa.edu (G.A.B.); 2Department of Pathology, University of Iowa, Iowa City, IA 52242, USA; zeb-zacharias@uiowa.edu; 3Interdisciplinary Graduate Program in Immunology, University of Iowa, Iowa City, IA 52242, USA; 4Department of Internal Medicine, University of Iowa, Iowa City, IA 52242, USA; 5Iowa City VA Health Care System, Iowa City, IA 52246, USA

**Keywords:** CD40, enveloped virus, macrophage, influenza A virus, Ebola virus, recombinant vesicular stomatitis virus, interferon gamma, IL-12, CD154, peritoneum, filovirus

## Abstract

Macrophages are critical in the pathogenesis of a diverse group of viral pathogens, both as targets of infection and for eliciting primary defense mechanisms. Our prior in vitro work identified that CD40 signaling in murine peritoneal macrophages protects against several RNA viruses by eliciting IL-12, which stimulates the production of interferon gamma (IFN-γ). Here, we examine the role of CD40 signaling in vivo. We show that CD40 signaling is a critical, but currently poorly appreciated, component of the innate immune response using two distinct infectious agents: mouse-adapted influenza A virus (IAV, PR8) and recombinant VSV encoding the Ebola virus glycoprotein (rVSV-EBOV GP). We find that stimulation of CD40 signaling decreases early IAV titers, whereas loss of CD40 elevated early titers and compromised lung function by day 3 of infection. Protection conferred by CD40 signaling against IAV is dependent on IFN-γ production, consistent with our in vitro studies. Using rVSV-EBOV GP that serves as a low-biocontainment model of filovirus infection, we demonstrate that macrophages are a CD40-expressing population critical for protection within the peritoneum and T-cells are the key source of CD40L (CD154). These experiments reveal the in vivo mechanisms by which CD40 signaling in macrophages regulates the early host responses to RNA virus infection and highlight how CD40 agonists currently under investigation for clinical use may function as a novel class of broad antiviral treatments.

## 1. Introduction

CD40 is a member of the tumor necrosis factor (TNF) receptor superfamily, a diverse group of signaling receptors characterized by their structural similarities that regulate a host of cellular processes. CD40 is expressed on a variety of different cell types, including antigen presenting cells (APCs) such as B-cells, dendritic cells (DCs), and macrophages (Mφs) and non-immune cells such as endothelial cells, epithelial cells, and fibroblasts [1]. CD40 interaction with its ligand CD40L (CD154) leads to an intracellular signaling cascade, resulting in pro-inflammatory responses and enhanced antigen presentation that are vital for the generation of robust cellular and humoral immunity [1]. Additionally, CD40 signaling is important in facilitating leukocyte adhesion, platelet activation, and wound healing, among other processes [1,2,3].

The signaling events downstream of CD40 ligation occur through a variety of TRAF-dependent mechanisms [4]. These include eliciting the canonical and non-canonical (IKK independent) NF-κB pathways, which leads to the production of pro-inflammatory cytokines such as TNF, IL-1β, IL-6, and IL-12, among others [4]. Other signaling cascades activated by CD40 include MAPK, PI3K/AKT, and JAK/STAT signaling which perform a variety of biological functions, such as stimulating the production and secretion of pro-inflammatory cytokines, facilitating cell differentiation, and regulating key cellular processes such as proliferation and apoptosis [4].

Given the importance of CD40 in coordinating adaptive immune responses, it has been well studied in the context of the activation and recruitment of B- and T-cells that are critical for controlling a variety of chronic bacterial and parasitic infections [5,6,7,8]. By contrast, the roles of CD40 in regulating innate immune cell activation and function and protection from acute viral infections are less well-characterized. However, recent work has begun to investigate acute innate immune responses that are triggered by CD40 signaling [9,10]. We reported that the ex vivo stimulation of CD40 signaling in resident peritoneal Mφs (pMφs) results in rapid and profound antiviral activity through the production of interferon gamma (IFN-γ). Our studies in pMφs found that CD40 signaling inhibits a variety of different RNA viruses as early as 12 h following infection [9]. *CD40*^−/−^ mice were significantly more susceptible to infection with a recombinant vesicular stomatitis virus (VSV) that encodes Ebola virus glycoprotein (rVSV-EBOV GP). These findings underscore a critical role for CD40 in amplifying innate antiviral immune responses.

Here, we investigate the in vivo requirements for CD40 signaling and the associated downstream pathways required for the innate immune control of two distinct murine models of virus infection and disease. Relative to WT control mice, *CD40^−/−^* mice challenged intranasally with influenza A virus (IAV) (PR8) exhibited elevated viral lung titers as early as 3 days following infection and decreased lung function as measured by whole-body plethysmography, as well as increased mortality. Similarly, using intraperitoneal rVSV-EBOV GP infection of type I interferon receptor deficient (*Ifnar^−/−^*) mice, we demonstrate that the administration of biologics that stimulate CD40 signaling protected mice against virus-induced mortality. We also tested the mechanism of CD40-mediated antiviral signaling and found that T-cell-Mφ interactions led to IL-12 production, which in turn promoted IFN-γ-dependent protection from infection. Together, this study identifies the in vivo mechanisms by which CD40 signaling in macrophages regulates the early host response to RNA virus infection and highlights that the CD40 agonists currently in clinical trial may serve as a novel class of broad antiviral treatments.

## 2. Materials and Methods

### 2.1. Ethics Statement

This study was conducted in accordance with the Animal Welfare Act and the recommendations in the Guide for the Care and Use of Laboratory Animals of the National Institutes of Health (University of Iowa (UI) Institutional Assurance Number: #A3021-01). All animal procedures were designed to minimize animal discomfort and were approved by the UI Institutional Animal Care and Use Committee (IACUC) which oversees the administration of the IACUC protocols, and the study was performed in accordance with the IACUC guidelines (protocols #8011280 Filovirus glycoprotein/cellular protein interactions and #1031280 virus glycoprotein/cellular protein interactions).

### 2.2. Mice

WT C57BL/6J mice were purchased from Jackson Labs (strain #000664). C57BL/6 *CD40^−/−^* mice were bred in house as previously described [9,11]. C57BL/6J *Ifnar^−/−^* mice were bred in house and a kind gift from Dr. John Harty (University of Iowa, Iowa City, IA, USA). C57BL/6J *CD40^−/−^Ifnar^−/−^* mice and C57BL/6J *Ifnar^−/−^ Ifngr1^−/−^* mice were crossed and maintained at the University of Iowa as previously described [9]. The number, age, and sex of the mice used for individual experiments are described in the figure legends.

### 2.3. Adoptive Transfers

Peritoneal macrophages were harvested from C57BL/6J *Ifnar^−/−^* mice (varying ages) as previously described [9]. To isolate peritoneal cells, the peritoneal cavity was lavaged with 10 mL cold RPMI (10% FBS, 1% pen/strep), and the lavage fluid was centrifuged, RBCs were lysed using ammonium-chloride-based lysis buffer, and cells were pelleted and resuspended in macrophage complete media (RPMI, 10%FBS, 1% sodium pyruvate, 1% L-glutamine, 1% pen/strep). The cells were cultured overnight to allow for macrophage adherence and then washed 3× to remove non-adherent cells. Cells were gently lifted (Versene, 10 min, on ice), washed, and resuspended in macrophage complete media. A total of 1 × 10^6^ macrophages in 100 µL of media were transferred by an intraperitoneal injection into recipient 6–8-week-old female *Ifnar^−/−^* or *Ifnar^−/−^CD40^−/−^* C57BL/6J mice and the mice were allowed to rest for 24 h prior to the rVSV-EBOV GP challenge. These studies were performed as two independent experiments and data were pooled.

### 2.4. Bone Marrow Transplants

Six–eight-week-old female C57BL/6J *Ifnar^−/−^* mice underwent whole-body irradiation (475 rads over ~5 min) delivered by an X-ray irradiator. Mice were allowed to rest for 4 h before undergoing a second round of irradiation (475 rads, ~5 min). In parallel, bone marrow was harvested from male and female *Ifnar^−/−^* and *Ifnar^−/−^CD40^−/−^* C57BL/6J mice as previously described [12]. Briefly, bone marrow was collected from the femurs of mice by flushing the exposed lumen with cold RPMI (10% FBS, 1% pen/strep) using a 27-gauge needle and attached a 10 mL syringe. Cells were filtered through a 70-micron cell strainer, RBCs were lysed with ammonium chloride containing lysis buffer, and cells were resuspended in RPMI (10% FBS, 1% pen/strep) and stored on ice until the time of transfer. Following irradiation, 1 × 10^6^ hematopoietic cells in 200 µL of media from donor mice were transferred to recipients via a tail vein injection. Following this, transfer mice were fed mouse chow containing Uniprim for 14 days before being switched to regular chow. Six weeks post-irradiation, the mice were used for downstream applications. Of note, it is our observation that *Ifnar^−/−^* mice are extraordinarily sensitive to the levels of irradiation necessary to deplete the marrow. For these experiments, excess mice were utilized to account for anticipated deaths following transfer. This resulted in an uneven number of mice in the four treatment groups.

### 2.5. Antibodies, Inhibitors, and Cytokines

Agonistic mouse anti-CD40 (clone FGK4.5/FGK45, BioXCell, Lebanon, NH, USA), anti-CD154/CD40L (MR-1, BioXCell), anti-NK1.1 (PK136, BioXCell), anti-CD19 (1D3, BioXCell), anti-IL12 (R1-5D9, BioXCell), and anti-IFNγ (XMG1.2, BioXCell) antibodies were diluted to 200 µg in PBS and administered 24 h prior to viral challenge. Anti-CD4 (YTS 177, BioXCell) and anti-CD8 (2.43, BioXCell) antibodies were diluted to 200 µg/100 µL in PBS, pooled to a total of 400 µg (200 µL), and administered 24 h prior to downstream use. Isotype IgG controls (BioXCell) were used at equivalent doses in all experiments. All antibodies were administered in a final volume of 100–200 µL via intraperitoneal (i.p.) injection. Cytokines used in this study include IL-12 (StemCell Technologies, Vancouver, CA, USA, catalog #78028.1) and IFN-γ (StemCell Technologies, catalog #78020.1), which were given at 5 µg in 100 µL of PBS via i.p. injection 24 h prior to viral challenge. For the disruption of IL-12 production, the inhibitor apilimod (MedChemExpress, Monmouth Junction, NJ, USA, catalog #HY-14644) was administered via i.p. injection at a concentration of 2.5 mg/kg. The drug was administered on day −1, day 0, and day 1 post-infection with rVSV-EBOV GP.

### 2.6. rVSV-EBOV GP Infections

Viral stocks were prepared as previously described [13]. Briefly, we used infectious, recombinant vesicular stomatitis virus bearing the glycoprotein from EBOV (Mayinga) and a GFP reporter in place of the native VSV-G. The virus was propagated by infecting Vero cells (MOI 0.1) and collecting supernatants which were filtered through a 0.45 µm filter and purified via ultra-centrifugation (28,000× *g*, 4 °C, 2 h) through a sucrose cushion. Stocks were then treated using an endotoxin removal kit (Detoxi-Gel Endotoxin Removing Gel, Thermo Fisher Scientific, Waltham, MA, USA) before being aliquoted and stored at −80 °C. Viral titers were determined according to the Reed–Muench method and reported as TCID50/mL [14].

For each stock, the LD_50_ was determined by the survival curves obtained by administering serial dilutions of the stock intraperitoneally (i.p.) to male and female C57BL/6J *Ifnar^−/−^* mice. Of note, female mice require a ~1/2 log increase in viral dose to achieve equivalent lethality. Studies were performed using one of two different doses of virus: a “sublethal dose”, which is chosen when 0–50% lethality was desired, and a “lethal” dose, which was chosen when >50% lethality was desired. Typically, a lethal dose in females was found to be 5 × 10^2^ infectious units (iu) (as assessed in a TCID_50_ assay performed in Vero cells), and a sublethal dose was found to be 7 × 10^1^ iu.

### 2.7. Influenza A Virus Infections

Mouse-adapted influenza H1N1 A/Puerto Rico/8/34 (PR8) was generated and prepared from laboratory stocks as previously described [15]. Six to ten-week-old male and female mice were infected intranasally with a LD_50_ (500 iu) of PR8 by administering virus in 50 µL of PBS to the nostrils of mice under isoflurane anesthesia and allowing for inhalation. While the initial weights of the mice were variable, all direct comparisons were made between age-matched animals with similar starting weights. Airway resistance was determined by measuring enhanced pause (Penh) and minute volume using whole-body plethysmography (WBP) (Buxco FinePointe, DSI, New Brighton, MN, USA).

### 2.8. Organ Harvest and qRT-PCR

Mice were perfused through the left ventricle with 5 mL cold sterile PBS prior to being euthanized by rapid cervical dislocation. Mice were under anesthesia with inhaled isoflurane throughout the duration of the procedure in accordance with our IACUC protocol. Organs were harvested and snap-frozen in liquid nitrogen to preserve tissue and virus prior to RNA isolation. RNA from organs was isolated using the TRIzol reagent from Thermo Fisher Scientific (Waltham, MA, USA) in accordance with the manufacturer’s specifications. Organs were dissociated for 1 min in 1ml TRIzol using the gentleMACS tissue dissociation system with associated gentleMACS M tubes (Miltenyi Biotec, Bergishch Gladbach, Germany). A total of 1 µg of RNA was subsequently converted to cDNA with the High-Capacity cDNA RevTrans Kit (#4368814) from Thermo Fisher Scientific. Quantitative PCR was performed using POWER SYBR Green Master Mix (#4367659) (Thermo Fisher Scientific) according to the manufacturer’s instructions and utilizing a 7300 real time PCR machine (Thermo Fisher Scientific). Twenty nanograms of cDNA was used in each well. All primers used were generated using NCBI’s “pick primers” tool. The primers used are as follows: cyclophilin forward 5′-GCT GGA CCA AAC ACA AAC GG-3′, cyclophilin reverse 5′-ATG CTT GCC ATC CAG CCA TT-3′, IAV forward 5′-CTT CTA ACC GAG GTC GAA AC-3′, and IAV reverse 5′-CGT CTA CGC TGC AGT CCT C-3′.

### 2.9. Data Presentation and Analysis

Experiments were performed a minimum of three independent times (except when indicated in the figure legend) and are shown with the error expressed as the mean with uncertainty displayed as the standard error of the mean. Statistical significance was determined by Student’s two-tailed *t*-test (alpha = 0.05) or Log-rank (Mantel–Cox) test as indicated in the figure legends. All figures were prepared using GraphPad Prism V 9.4 (Dotmatics, Bishop’s Stortford, UK), and statistical analyses were performed using Prism’s default settings.

## 3. Results

### 3.1. CD40 Signaling Plays a Critical Protective Role at Early Timepoints after Infection with Influenza A Virus

We previously showed that CD40 signaling exerts a rapid and broadly antiviral effect in cultured primary pMφs and reduces virus load at 24 h following the intraperitoneal (i.p.) infection of mice with recombinant vesicular stomatitis virus (VSV) bearing the Zaire Ebola virus (EBOV) glycoprotein (rVSV-EBOV GP) [9]. To build upon these observations and determine if our findings were applicable to a wider array of viral infections, we investigated whether CD40 signaling exerted a similar protective effect within the lung during acute influenza A virus (IAV) infection.

Our earlier studies demonstrated that the stimulation of CD40 signaling protects pMφs ex vivo from infection with a mouse-adapted strain of IAV (Puerto Rico/8/34 or PR8) [9]. Here, we extend this by examining the effect of CD40 signaling in WT and *CD40*^−/−^ mice infected with IAV PR8 intranasally (i.n.). *CD40*^−/−^ mice were more susceptible to a ~LD_50_ dose of virus (Figure 1A), with a greater number of mice succumbing to infection at earlier timepoints. Additionally, WT mice that were treated with an agonistic CD40 antibody 24 h prior to infection exhibited decrease susceptibility to viral challenge (Figure 1A). As our in vitro studies suggested that the protective effect of CD40 signaling is due to IFN-γ production, we treated WT and *CD40*^−/−^ mice with IFN-γ or PBS 24 h prior to i.n. challenge with PR8 and found that IFN-γ decreased mortality in both groups (Figure 1B), with IFN-γ conferring more robust protection in the *CD40*^−/−^ group given their relatively increased susceptibility at the baseline. Furthermore, our in vitro studies identified IL-12 as a key intermediary to IFN-γ production [9]. To evaluate the role of IL-12 and IFN-γ in our intranasal infection model, we treated mice with an antagonistic IL-12 antibody or an antagonistic IFN-γ antibody prior to challenge with PR8 and found that the administration of blocking antibodies trended towards enhanced disease compared to untreated controls, although the trends were not statistically significant (Figure 1C). In contrast to our rVSV-EBOV GP model, where mice die within 3–5 days, the disease course of PR8 infections is more prolonged, with infected mice succumbing from 8 to 11 days [16]. Thus, our IAV survival curve findings are likely impacted not only by innate responses but also by the well-characterized role of CD40 in regulating adaptive immune processes that are elicited in this time frame [17]. Additionally, early innate immune cell functions are known to lead to the induction of protective adaptive immunity [18].

To directly investigate PR8-induced disease prior to the induction of adaptive immune responses, we assessed early PR8-associated morbidity by measuring lung function using whole-body plethysmography (Buxco boxes) [19,20,21]. WT and *CD40*^−/−^ mice were infected i.n. with a sublethal dose of PR8, and PenH scores (a measure of respiratory pause that correlates with airway resistance) and minute volume (a measure of airflow in the lungs) were measured daily. While both strains of mice had elevated PenH scores and decreased minute volume as PR8 infection progressed, the effects were significantly more pronounced and occurred earlier in the *CD40*^−/−^ mice (Figure 1D–E, Appendix A), indicative of decreased lung function within three days of infection. These data provide evidence of enhanced disease in CD40-deficient animals during acute viral infection prior to the development of adaptive immune responses. To evaluate whether this decrease in function was associated with increased viral burden in the lung, we harvested lungs from infected mice and found significantly increased viral loads by qRT-PCR at day 3 (Figure 1F). We also note that by 6 dpi, there were no differences in viral load between infected *CD40*^−/−^ and WT mice. This is consistent with our prior reported data with the rVSV-EBOV GP model where early virus loads (24 h) were elevated in *CD40*^−/−^ mice compared to CD40-competent mice, but viral loads did not differ by 48 h of infection [9]. These data indicate that loss of CD40 signaling decreased the host control of influenza A virus infection at time points that precede the arrival of adaptive immunity in the lungs, and this was associated with enhanced early pathology and greater morbidity and mortality.

### 3.2. CD40 Signaling in Peritoneal Macrophages Exerts an Antiviral Effect in Mice

Given that CD40 plays a critical role in protecting mice during PR8 infection and that we saw a similar effect previously with rVSV-EBOV GP [9], we sought to evaluate the mechanism by which this occurs in vivo. To do this, we turned to i.p. infections with rVSV-EBOV-GP in *Ifnar^−/−^* mice, a model of infection utilizing a virus that targets Mφs, but not lymphocytes or other cell types within the peritoneal compartment [22,23,24]. An advantage to this approach is that infection outcomes are tightly linked with the dose of virus given [9], providing a range of outcomes from no disease to predictable mortality within 3–5 dpi. Further, the ability to elicit severe pathology within a short time allows insights into in vivo innate immune orchestration and function, leading to protection in a time frame insufficient for significant (and complicating) adaptive responses. Our use of *Ifnar^−/−^* mice additionally eliminates the antiviral impact of type I IFNs, allowing the studies to focus on specific pathways stimulated by CD40 signaling.

Given that *Ifnar^−/−^CD40^−/−^* mice are more susceptible to rVSV-EBOV GP infection compared to CD40-sufficient *Ifnar^−/−^* mice [9], we first tested whether the stimulation of CD40 would protect *Ifnar^−/−^* mice from a lethal challenge. To do this, we administered agonistic CD40 antibody or IgG to CD40-sufficient mice 24 h prior to a lethal challenge with rVSV-EBOV GP. We found that mice treated with agonistic CD40 antibody were significantly less likely to succumb to disease, and that death was delayed compared to unstimulated animals (Figure 2A). In contrast, blocking CD40 signaling by the administration of an anti-CD154 (CD40 ligand) antibody resulted in markedly enhanced lethality in mice challenged with a sublethal dose of virus. This demonstrated that the acute blockade of CD40 signaling or a germline deficiency of CD40 results in similar effects on early antiviral responses (Figure 2B, reference [9]).

Our previous ex vivo experiments utilized resident pMφs as they express CD40, which is readily obtained from the peritoneal cavity, and are a key target of our model virus [9,23]. However, it is important to note that CD40 expression is not limited to Mφs or even cells of the hematopoietic compartment. CD40 is also found on a variety of other cell types, including myocytes, fibroblasts, and epithelial cells [1]. To determine whether hematopoietic cells, and specifically pMφs, expressing CD40 are protective against rVSV-EBOV GP infection, we performed bone marrow transplants as well as adoptive transfers of pMφs. For bone marrow transfer studies, we harvested bone marrow from *Ifnar^−/−^* or *Ifnar^−/−^CD40^−/−^* mice and transferred these cells to irradiated *Ifnar^−/−^* recipients. When mice were challenged with a sublethal dose of rVSV-EBOV GP following engraftment, we found that mortality was significantly increased in mice receiving *CD40*^−/−^ marrow but not in those receiving marrow from CD40-sufficient donors (Figure 2C). Furthermore, the mice receiving *CD40*^−/−^ marrow were indistinguishable from *Ifnar^−/−^ CD40^−/−^* controls, suggesting that CD40 on hematopoietic cells is vital for mediating antiviral signaling. To directly test the role of CD40 on pMφs, we harvested peritoneal cells from *Ifnar^−/−^* or *Ifnar^−/−^CD40^−/−^* mice, allowed pMφs to adhere in tissue culture, removed non-adherent cells, and transferred pMφs to *Ifnar^−/−^CD40^−/−^* recipients. It is important to note that resident pMφs were not depleted as we have previously found this process to induce inflammation, resulting in pMφ polarization that interferes with rVSV-EBOV GP replication, thereby biasing our results. When CD40-sufficient pMφs were introduced to the peritoneal cavity of *Ifnar^−/−^CD40^−/−^* mice, we found that the presence of CD40-expressing Mφs was sufficient to protect mice from succumbing to infection (Figure 2D). In total, these studies support that CD40 expression by pMφs is essential for optimal protection against rVSV-EBOV GP challenge.

### 3.3. CD40 Signaling Exerts an Antiviral Effect in Mice through IL-12 and IFN-γ Production

Given our cell culture findings establishing a critical role for both IFN-γ and Mφ expression of CD40 for protection against acute virus infection [9], we interrogated whether IL-12 was functionally linked to IFN-γ production and protection in vivo. To investigate the role of IL-12 in vivo, we treated *Ifnar^−/−^* and *Ifnar^−/−^CD40^−/−^* mice with recombinant IL-12 or PBS and infected mice with a lethal dose of rVSV-EBOV GP 24 h later. Consistent with our ex vivo studies [9], we found that recombinant IL-12 protected mice from death regardless of the presence of CD40 (Figure 3A,B). To directly ask whether IL-12 is essential for protection in our in vivo virus challenge model, we blocked its production using apilimod, a PIKfyve inhibitor that blocks both IL-12 and IL-23 production in vitro and in vivo [25,26,27,28]. We found that *Ifnar^−/−^* mice exhibited increased mortality following challenge with a sublethal dose of rVSV-EBOV GP when administered apilimod from day −1 to day 1 post-infection. However, we did not observe a change in mortality in *Ifnar^−/−^CD40^−/−^* mice administered apilimod (Figure 3C,D). This provides independent evidence that IL-12 production is an important downstream mediator of CD40 signaling. To evaluate the key role of IFN-γ in our model, we treated *Ifnar^−/−^ Ifngr^−/−^* mice with either a CD40 agonist or recombinant IL-12 24 h prior to challenge with rVSV-EBOV GP and found that neither stimulus protected mice in the absence of IFN-γ signaling (Figure 3E,F). These studies show that CD40 on the surface of pMφs protects mice against acute RNA virus challenge by stimulating IL-12 production, that in turn drives IFN-γ production.

### 3.4. Peritoneal T-lymphocytes Stimulate CD40 Signaling in pMφs

The studies outlined above stimulated CD40 signaling with an agonistic antibody; however, physiologically cell surface or soluble CD154 interacts with CD40 to elicit those signaling events [1]. We previously demonstrated that the depletion of peritoneal CD3+ T-cells in ex vivo cultures increased titers of infectious rVSV-EBOV GP and that peritoneal T-cells from 4 to 6-week-old specific pathogen-free mice express detectable levels of surface CD154, suggesting it may be peritoneal T-cells that provide the CD154 ligand [9]. Thus, we next investigated whether the depletion of CD4+ and CD8+ cells elicited earlier mortality and more severe pathology following sublethal challenge with rVSV-EBOV GP. CD4^+^ and CD8^+^ T-cell populations were systemically depleted by administering antibodies to mice 24 h prior to infection with rVSV-EBOV GP. We found that T-cell depletion accelerated the time-to-death and increased overall virus-elicited mortality in *Ifnar^−/−^* mice (Figure 4A). By contrast, T-cell depletion did not increase either the rate or overall mortality of *Ifnar^−/−^CD40^−/−^* mice (Figure 4B), consistent with the provision of CD154 by T-cells in vivo. However, because the antiviral effects of CD40 signaling are dependent on IFN-γ production, T-cells are one of several populations of peritoneal cells capable of IL-12-induced IFN-γ production [29], meaning it was possible that T-cell depletion enhanced disease due to the loss of IFN-γ production and not the loss of CD154. To distinguish between these possibilities, we depleted T-cells in *Ifnar^−/−^* mice and administered agonistic CD40 antibody to bypass the need for CD154 (Figure 4C). Using this approach, we found that CD40 agonist protected mice regardless of the presence or absence of T-cells. This suggests that IL-12-mediated production of IFN-γ, at least in part, can occur through the activation and function of other populations that may include NK cells and ILCs.

Collectively, these findings support that CD40-CD154 interactions between T-cells and pMφs initiate a signaling cascade that results in IFN-γ production, conferring protection against RNA virus infection in mice. These critical circuits are engaged upon virus infection, which supports that CD40 signaling plays a critical role in coordinating innate immune responses.

## 4. Discussion

Here, we show the critical role of macrophage CD40 signaling in the control of virus infection and early pathology associated with negative-strand RNA virus infection of mice. CD40 signaling is appreciated to contribute to a wide variety of host immune responses, including the production of robust T-cell-dependent antibody responses, dendritic cell licensing, and macrophage activation [1]. These host responses to this signaling cascade are documented to control bacterial and parasitic infection as well as herpesvirus reactivation [30,31,32,33], but the contribution of innate CD40 signaling during acute virus infection and the associated pathology have not been extensively studied. Our prior study demonstrated that loss of CD40 in purified pMφs enhances the virus load of VSV, Sindbis virus, IAV, EBOV, and Ross River virus at 24 h of infection and even as early as 12 h in in vivo studies with rVSV-EBOV GP [9]. Conversely, agonistic CD40 antibody stimulation of the pathway suppresses infection by these viruses. Our current studies reveal the in vivo mechanisms by which innate CD40 signaling functions to control IAV and rVSV-EBOV GP infection, information that may be broadly relevant for the development of novel strategies for anti-viral therapy.

Loss of CD40 signaling enhances IAV infection and exacerbates respiratory distress at timepoints that are prior to the production of significant adaptive immune responses. However, this effect is transient as IAV titers and measures of breathing difficulty begin to converge in CD40-sufficient and -deficient mice by day 6 of IAV infection. Loss of CD40 signaling led to greater mortality beginning on day 7 and is likely multifactorial, with enhanced mortality resulting from an absence of viral control early during infection as well as deficits in the recruitment and activation of adaptive immunity at later times [17]. Hence, early elevated viral load and pathology is likely working in concert with more traditional deficits in adaptive immune responses that would be anticipated in *CD40*^−/−^ mice, ultimately leading to worse outcomes [11,34].

During IAV or rVSV-EBOV GP infections, we find that mice are protected by the administration of exogenous IFN-γ (and IL-12 in the case of rVSV-EBOV GP) regardless of whether the CD40 signaling axis is intact. However, the effect is more pronounced when CD40 signaling is absent (Figure 1, [9]). This observation is most readily explained by physiological levels of CD40 signaling inducing quantitatively less endogenous IL-12 or IFN-γ than that we administered exogenously. Alternatively, the timing of cytokine exposure (prior to infection versus produced in response to infection) may dictate the magnitude of the response. Regardless, our data indicate that these cytokines are key intermediate steps in the protection conferred by CD40 against these RNA viruses. While it is appreciated that IFN-γ is protective against EBOV and rVSV-EBOV GP [24,35], the protective role of IFN-γ during IAV infection is less clear [36,37,38]. These studies contribute insights specifically into cellular pathways, contributing to innate protection against IAV. In a broader context, we provide a novel conceptual framework for understanding an early innate immune pathway that controls acute virus infection independent of type I IFN signaling.

Our studies demonstrated that CD4+ or CD8+ T-cells provide the CD154 ligand which interacts with CD40 on pMφ. Consistent with this, we had previously reported the detection of modest levels of CD154 on lymphocytes present in the peritoneum of uninfected mice [9]. Presumably, rVSV-EBOV GP infection would lead to the enhancement of CD154 present on these cells as infections have previously been reported to elevate CD154 on T-cells [39]. Furthermore, tissue-resident CD4+ T-cells present in the spleens of pathogen-free mice have been shown to contain preformed CD154 within their lysosomal compartment, and CD154 is rapidly trafficked to the plasma membrane upon T-cell activation [40]. Additionally, effector memory T-cell CD154 interactions with CD40+ myeloid cells can elicit the innate immune activation and production of IL-12 and other proinflammatory cytokines [41]. Hence, the presence of memory T-cells in the peritoneum and elicited by prior stimuli likely serve as the source of CD154, thereby providing an avenue for innate immune protection when a virus acutely infects peritoneal cells, such as pMφs that express CD40. What remains to be understood is the precise mechanism by which this signaling is initiated and whether the cascade of events is similar in different organs.

The results of bone marrow and peritoneal cell transfers in our rVSV-EBOV GP model provide evidence that hematopoietic cells, and specifically pMφs, are a key CD154-responsive population in the context of peritoneal infection (Figure 2C,D), with pMφs being responsible for CD40 signaling early in infection. Given our previous work showing that CD40 signaling generates an rVSV-EBOV GP-resistant, M1-like phenotype in pMφs [9], we contend that these cells are also the key IFN-γ responsive population, and thus play a dual role in recognizing/stimulating innate immune responses and responding to viral infection. While it remains to be seen if similar events contribute to innate responses to IAV infection, it is notable that alveolar macrophages are critical mediators of the antiviral response to IAV [42].

A remaining knowledge gap is the identification of the cells that elicit IFN-γ in response to IL-12. Of the cells present in the peritoneal cavity, T-lymphocytes, NK cells, and innate lymphoid cells are all implicated in IFN-γ production. Thus, there may be functional redundancies. Additionally, IFN-γ production does not necessarily need to occur in the peritoneal cavity, thus a more expansive investigation of additional lymphoid populations is warranted. Future work analyzing the response to infection at a single-cell resolution would allow for a dissection of these events. Regardless of the source of IFN-γ production, the observation that CD40 signaling plays a key protective role against diverse viral pathogens has important implications. Not only are agonists of CD40 increasingly clinically available, but a clearer understanding of the signaling events that occur early during acute viral infection may open up the door towards the development of needed antiviral therapies.

## Figures and Tables

**Figure 1 viruses-15-01353-f001:**
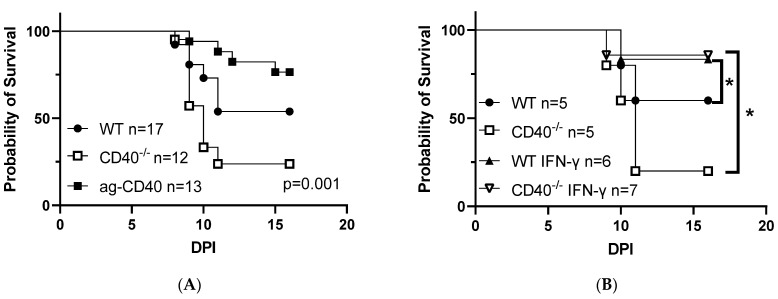
Loss of CD40 increases susceptibility of mice to IAV and enhances morbidity at early time points following infection. (**A**) Male and female WT, WT stimulated with 200 µg of agonistic CD40 antibody, or *CD40^−/−^* mice were infected intranasally with IAV PR8 at an LD_50_. Survival was monitored, and statistical analysis using the Log-rank test was performed. (**B**) Male and female WT or *CD40^−/−^* mice were treated with PBS or 5 µg of IFN-γ. Twenty-four hours following treatment, mice were infected intranasally with IAV PR8. Survival was monitored and statistical analysis using Log-rank test was performed. (**C**) Male and female WT mice or WT mice treated with 200 µg of antagonistic (blocking) IL-12 or IFN-γ monoclonal antibody were infected intranasally with IAV PR8 at an LD_50_. Survival was monitored, and statistical analysis using the Log-rank test was performed. (**D**–**F**) Male and female WT or *CD40^−/−^* mice were infected intranasally with IAV PR8 at an LD_50_. Morbidity was scored using whole-body plethysmography to measure two metrics to quantify breathing difficulty, PenH scores (**D**), and minute volume (**E**). Lungs from a subset of mice were harvested at the indicated time point and viral loads were quantified by qRT-PCR (**F**). Statistical analyses were performed using two-way ANOVA (**D**–**E**) or Student’s *t*-test (**F**) with * indicating *p* < 0.05.

**Figure 2 viruses-15-01353-f002:**
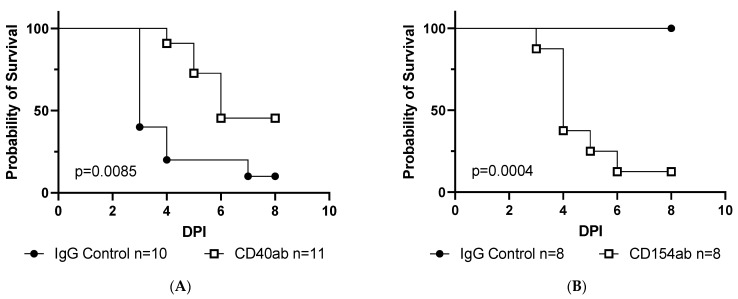
In vivo stimulation/blockade of CD40 on peritoneal macrophages significantly impacts rVSV/EBOV GP infection. (**A**,**B**) Male and female C57BL/6J *Ifnar^−/−^* mice (6–8 weeks old) were treated with 200 µg of the agonistic CD40 monoclonal antibody, FGK4.5/FGK45 (**A**), or antagonistic (blocking) CD154 antibody, MR-1 (**B**), or control IgG. Twenty-four hours after administration, mice were infected with either a lethal (**A**) or sub-lethal (**B**) dose of rVSV-EBOV GP. (**C**) Bone marrow transfers. Bone marrow was harvested from *Ifnar^−/−^* (CD40-sufficient) or *Ifnar^−/−^CD40^−/−^* mice and 1 × 10^6^ cells were transferred to recently irradiated *Ifnar^−/−^* mice. Mice, along with control *Ifnar^−/−^* or *Ifnar^−/−^CD40^−/−^* mice, were challenged on day 42 post-transfer with rVSV/EBOV-GP at an LD_50_. (**D**) Adoptive transfer of peritoneal macrophages. Peritoneal cells were harvested from *Ifnar^−/−^* mice and allowed to adhere to tissue culture plates for 24 h. Non-adherent cells were removed, and 1 × 10^6^ cells were transferred to *Ifnar^−/−^CD40^−/−^* mice via intraperitoneal injection. (**A**–**D**) Mice were monitored for survival, and statistical significance was determined by the Log-rank test.

**Figure 3 viruses-15-01353-f003:**
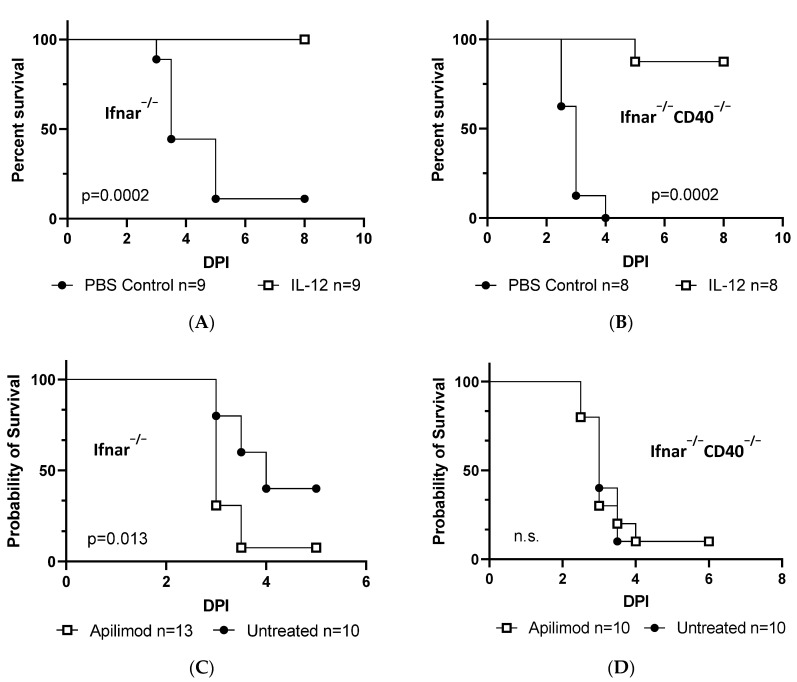
CD40-mediated protection from rVSV/EBOV-GP occurs in resident peritoneal macrophages in an IL-12-dependent manner; (**A**,**B**) 6–8-week-old C57BL/6 male and female *Ifnar^−/−^* (**A**) or *Ifnar^−/−^CD40^−/−^* (**B**) mice were treated with PBS or 5 µg of recombinant IL-12. Twenty-four hours following treatment, mice were challenged with a lethal dose of rVSV-EBOV GP. (**C**,**D**) Six–eight-week-old male and female *Ifnar^−/−^* (**C**) or *Ifnar^−/−^CD40^−/−^* (**D**) mice were treated with PBS or the IL-12 inhibitor, apilimod, at a concentration of 2.5 mg/kg. Twenty-four hours following treatment, mice were challenged with a lethal dose of rVSV-EBOV GP. (**E**,**F**) Six–eight-week-old male and female *Ifnar^−/−^Ifngr^−/−^* mice were treated with 200 µg of control IgG or agonistic CD40 antibody, FGK4.5/FGK45 (**E**), PBS, or 5 µg of IL-12 (**F**). (**A**–**F**) Survival was monitored, and statistical significance was calculated by the Log-rank test.

**Figure 4 viruses-15-01353-f004:**
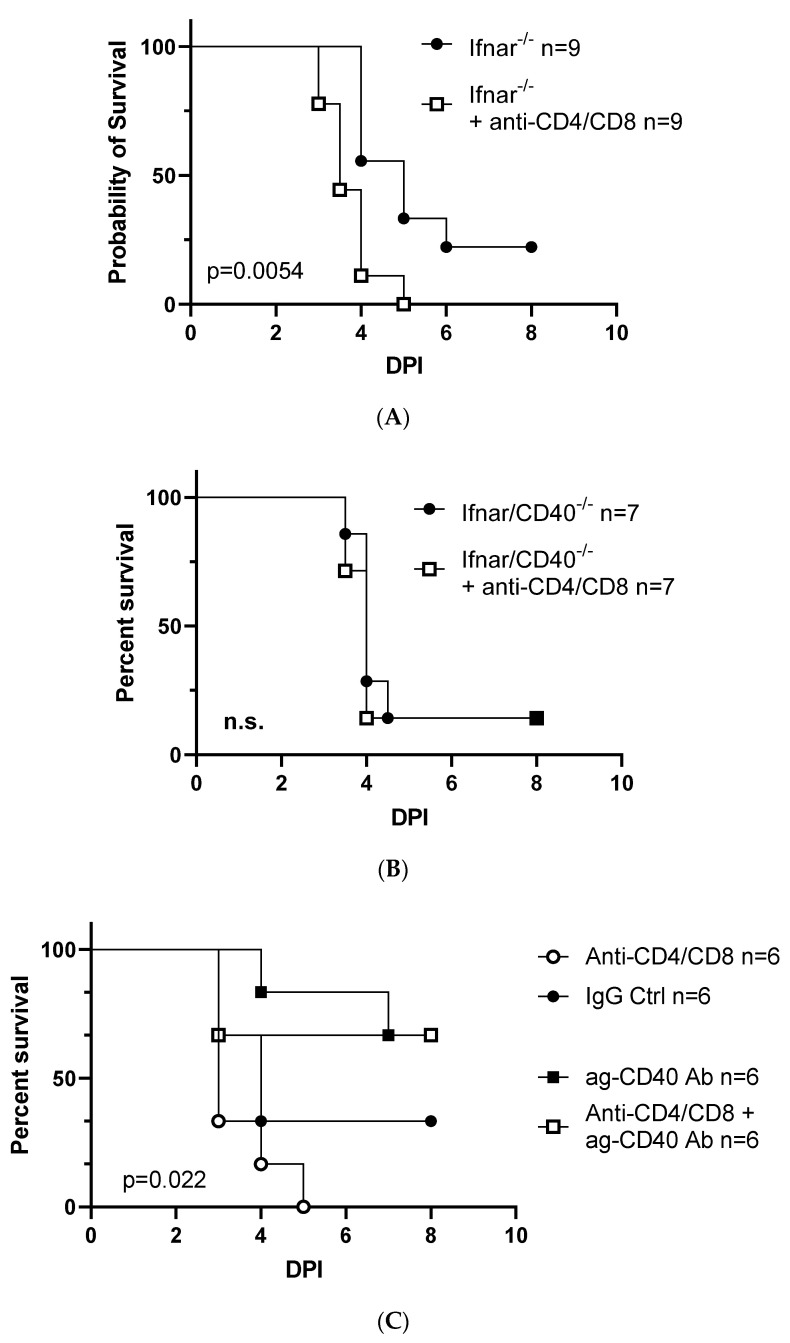
T-lymphocytes are a critical source of CD154 but are not essential for the protection conferred by downstream cytokines; (**A**–**C**) T-lymphocyte depletion studies. Six–eight-week-old male and female *Ifnar^−/−^* or *Ifnar^−/−^CD40^−/−^* mice were given 200 µg of agonistic CD40 antibody, depleting antibodies to CD4 and CD8, or isotype control IgG, as noted in each panel. Mice were subsequently infected with a sublethal dose of rVSV-EBOV GP, survival was monitored, and statistical significance was determined by Log-rank test. *Ifnar^−/−^* mice (**A**) or *Ifnar^−/−^CD40^−/−^* mice (**B**) received IgG or depleting CD4 and CD8 antibodies 24 h prior to challenge. (**C**) *Ifnar*^−/−^ mice received IgG (black symbols) or depleting CD4 and CD8 antibodies (white symbols). Twenty-four hours after antibody administration, mice were given either IgG (top two groups) or agonistic CD40 antibody (bottom two groups). Twenty-four hours after the second round of antibody administration, mice were challenged i.p. with rVSV-EBOV GP.

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
