# Peer review of "CD40 Signaling in Mice Elicits a Broad Antiviral Response Early during Acute Infection with RNA Viruses"

_viruses, 2023, doi:10.3390/v15061353_

Round 1

Reviewer 1 Report

Summary of the work

The authors share here their findings on the involvement of the CD40-CD40L signaling pathway during infection with RNA viruses, in vivo in mouse models upon intranasal (local) and intraperitoneal challenges. This work follows on a study recently published by the same team which presents evidence of the importance of the CD40 pathway in vitro, in particular in resident peritoneal macrophages. In the current study, the authors first present the deleterious effects on the survival, viral control and physiological capacities, i.e. respiratory function as assessed using WBP, of animals deficient for CD40 or by activatin the CD40 receptor with agonistic antibody during a viral infection, and point out the importance of IFN-γ and IL-12 in the anti-viral response. First of all, all the work is based on in vivo experimentation, and there is no observational or mechanistic data at the cell of molecular level, from ex vivo analysis such as flow cytometry or coculture, that could have made the parallel with the observations on animals. Different strains of mice were used, deficient for CD40, IFN-αR and/or IFN-γR, as well as adoptive transfer and bone marrow transplantation, and the viral load was monitored ex vivo by qPCR. Moreover, the survival time in the intraperitoneal mouse model being very short, we would have loved to see the mechanistic data with the use of agonist or blocking antibodies in the slightly more physiological intranasal model, since the mounting of the cellular response is not not immediate but requires a few days, if not a few weeks for the humorla response.

Minor comments

  • I suggest authors use a less general title, which reflects the breadth of findings published here, e.g. “CD40 signaling in mice elicits a broad antiviral response early during acute infection with RNA viruses”.
  • In Figure 1B, is it the CD40-/- strain with IFN-γ added in the group represented by the open inverted triangle?
  • In Figure 1C–E, the mixed-design ANOVA test best suited in this situation, followed if necessary by pairwize tests. Moreover, it would be more judicious to present the data for each animal in figures 1C–D (perhaps in supplementary material?), because the SEM error bars do not make it possible to realize the extent of the effects.
  • Can you justify the absence of 2 animals in the CD40-/- group (n=10) in Figure 1D vs. the related data presented in the rest of Figure 1 (with n=12 in the CD40-/- group)?
  • In Figure 2C we cannot see the number of animals in the last group

Author Response

Response to Reviewer #1: Our responses are italicized

The authors share here their findings on the involvement of the CD40-CD40L signaling pathway during infection with RNA viruses, in vivo in mouse models upon intranasal (local) and intraperitoneal challenges. This work follows on a study recently published by the same team which presents evidence of the importance of the CD40 pathway in vitro, in particular in resident peritoneal macrophages. In the current study, the authors first present the deleterious effects on the survival, viral control and physiological capacities, i.e. respiratory function as assessed using WBP, of animals deficient for CD40 or by activatin the CD40 receptor with agonistic antibody during a viral infection, and point out the importance of IFN-γ and IL-12 in the anti-viral response. First of all, all the work is based on in vivo experimentation, and there is no observational or mechanistic data at the cell of molecular level, from ex vivo analysis such as flow cytometry or coculture, that could have made the parallel with the observations on animals. Different strains of mice were used, deficient for CD40, IFN-αR and/or IFN-γR, as well as adoptive transfer and bone marrow transplantation, and the viral load was monitored ex vivo by qPCR. Moreover, the survival time in the intraperitoneal mouse model being very short, we would have loved to see the mechanistic data with the use of agonist or blocking antibodies in the slightly more physiological intranasal model, since the mounting of the cellular response is not not immediate but requires a few days, if not a few weeks for the humorla response.

We agree that the inclusion of studies with inhibitory antibodies in the influenza model strengthens our conclusions. We have added these data as a new panel that is now Figure 1C. Blocking antibodies against IL-12 or IFN-γ resulted in a trend towards worse survival, but those trends are not statistically significant. These data additions are discussed in the Results ((Lines 215-220) and detailed information on these experiments is included in Materials and Methods (line 129-130) and figure legend (lines 253-255).

- I suggest authors use a less general title, which reflects the breadth of findings published here, e.g. “CD40 signaling in mice elicits a broad antiviral response early during acute infection with RNA viruses”.

Thank you for this suggestion, we agree that this title more accurately represents our findings and we have incorporated this change.

- In Figure 1B, is it the CD40-/- strain with IFN-γ added in the group represented by the open inverted triangle?

Thank you for catching this. Yes, the CD40-/- strain with IFN-γ added is represented by the open inverted triangle. We have modified this Figure to more clearly illustrate the use of the CD40-/- strain.

- In Figure 1C–E, the mixed-design ANOVA test best suited in this situation, followed if necessary by pairwize tests. Moreover, it would be more judicious to present the data for each animal in figures 1C–D (perhaps in supplementary material?), because the SEM error bars do not make it possible to realize the extent of the effects.

Thank you for this suggestion. We have incorporated mixed-design ANOVA for figures 1C-D (now Figures 1D-E). For Figure 1E (now 1F), we use pairwise tests to highlight the significance of the difference in titer at Day 3 post infection.

Additionally, we have now included the raw data for Figures 1C-D (now D-E). It is shown currently as a Table S1 as suggested.

- Can you justify the absence of 2 animals in the CD40-/- group (n=10) in Figure 1D vs. the related data presented in the rest of Figure 1 (with n=12 in the CD40-/- group)?

Thank you for noting this discrepancy. Two animals from the CD40-/- group were erroneously omitted from the original Figure. These have been added to the Figure and the statistics have been updated (but this did not result in changes in the statistical findings).

- In Figure 2C we cannot see the number of animals in the last group

Thank you for catching this. There was a formatting error that has now been corrected.

Reviewer 2 Report

Nice work that widens previous in vitro findings published by the same group (ref. 9 of the present manuscript).

Well designed, performed and well written. My only point is that probably a third in vivo model would be ideal to reinforce the message of the paper.

You could use one of viruses already used for your in vitro studies- or  mouse adapted coronavirus infection maybe?

I am aware that this is a relatively simple point, but it means a lot of lab work! Furthermore, If you use a coronavirus model I am keen that it could be a paper per se...

THE Editor(s) will help in choosing the best and most viable trajectory.

Author Response

Responses to Reviewer #2: our responses are italicized

- Nice work that widens previous in vitro findings published by the same group (ref. 9 of the present manuscript).

- Well designed, performed and well written. My only point is that probably a third in vivo model would be ideal to reinforce the message of the paper.

- You could use one of viruses already used for your in vitro studies- or mouse adapted coronavirus infection maybe?

- I am aware that this is a relatively simple point, but it means a lot of lab work! Furthermore, If you use a coronavirus model I am keen that it could be a paper per se...

Thank you for the suggestion. We agree that a third virus would add to the significance of this work and appreciate your understanding regarding the difficulty of including these data. Of the viruses we used in vitro in our previous study, VSV would be the simplest to incorporate as we have experience with the virus and could utilize our Ifnar-/- mice for in vivo work. However, we decided not to include this virus as it shares so much in common with rVSV/EBOV GP (just a broader tropism). Other viruses either lack a reliable mouse model or were beyond our area of expertise and not in use by collaborators. We agree that exploration of a mouse adapted coronavirus would be of high interest and will plan to explore this in future work.   

Reviewer 3 Report

The current study aims to evaluate the role of CD40 in the immune response to RNA viruses. 

- The design of the study and the aim of the study are clear, however, the authors should elaborate on the importance of their findings and what is the merits of their research.

- The authors could consider performing more in-vitro tests to attest to the roles of CD40 

- Add references to line 40 and line 49

-VSV abbreviation 

The English is ok

Author Response

Responses to Reviewer #3: our responses are italicized

- The current study aims to evaluate the role of CD40 in the immune response to RNA viruses.

- The design of the study and the aim of the study are clear, however, the authors should elaborate on the importance of their findings and what is the merits of their research.

Thank you for the suggestion. We thought we had highlighted the importance of our findings

- The authors could consider performing more in-vitro tests to attest to the roles of CD40

Thank you for this suggestion. Our earlier study did explore the role of CD40 in vitro (reference 9 in the manuscript) in a more mechanistic manner. Further work aimed at understanding transcriptional changes at single cell resolution is underway and will form the basis of additional in vitro and in vivo experiments.

- Add references to line 40 and line 49

Thank you for catching these omissions, these references have now been included.

-VSV abbreviation

This abbreviation is defined on line 64 of the manuscript.